# Impaired slow-wave sleep accounts for brain aging-related increases in anxiety
Eti Ben Simon [1,2] ✉, Vyoma D. Shah[1,2], Olivia Murillo[1,2], Zavecz Zsofia[1,2] & Matthew P. Walker[1,2,3] ✉

Aging doesn't just dull our memories; it destabilizes our emotions while further impairing sleep quantity and NREM sleep quality. Emotional dysregulation and anxiety symptoms in older adults accelerate their risk of cognitive decline and dementia, yet the underlying mechanisms remain largely unclear. In young adults, reductions in deep sleep, specifically the loss of slow wave activity (SWA) during non-REM sleep, impair the brain's ability to regulate anxiety overnight. This raises a testable hypothesis: Does age-related decline in SWA contribute to increased anxiety symptoms in older adults? We test this hypothesis in 61 cognitively healthy older adults (>65 y) experiencing varying levels of anxiety. Each participant underwent polysomnography-recorded sleep in the lab, followed by a structural MRI the next morning to assess atrophy in anxiety-sensitive brain regions. A subset of 24 participants was tracked longitudinally over 4 ± 2.02 years. The findings were consistent. Greater impairment in nighttime SWA predicted higher next-day anxiety in older adults, both at baseline and at follow-up. Brain imaging revealed the mechanism: atrophy in key emotion-processing regions was associated with reduced capacity to generate robust slow waves needed for overnight anxiety regulation. Critically, mediation analysis showed that impaired SWA fully accounted for the relationship between regional atrophy and disrupted overnight anxiety regulation. These findings suggest that even in the presence of age-related brain atrophy, intact SWA may preserve emotional stability by rescuing the brain's nightly emotional recalibration process, protecting mental health in aging populations.

Aging is not just a matter of cognitive decline but of mental health decline. Such mental health challenges are marked, prevalent, and significantly life-disabling[1–3]. In addition, the management and treatment of mental health in patients with degenerative dementias is often one of the most challenging features of abnormal aging for caregivers, and the most painful for family members to endure[4,5]. Of special relevance, the transition from normative aging to dementia, including mild cognitive impairment and Alzheimer's disease, is either foreshadowed by or followed by escalations in mental health conditions[6,7]. All of which is to say, understanding and addressing mental health in aging populations is not merely a peripheral concern, but a central aspect of ensuring quality of life and managing the broader impacts of cognitive decline.

Age-related mental health disruptions have been systematically grouped into a set of features, termed neuropsychiatric symptoms (NPS). Neuropsychiatric symptoms are observed in up to 90% of patients with dementia at some point during their illness[8], and at their core, encompass anxiety, apathy, depression, and sleep disruption[2]. These symptoms are relevant, even in their earliest appearance, since their emergence and

subsequent severity significantly predict (1) an earlier entry into nursing home care and loss of independence[3,7], (2) greater functional impairment including difficulties with basic daily activities[9], (3) worse quality of life[10], (4) an accelerated progression to late-stage dementia[1], and (5) a significantly escalating burden of Alzheimer's disease (AD) pathology and brain atrophy[11,12].

Across the neuropsychiatric symptoms, anxiety and anxiety disorders are the most common mental illness in adults aged 60 or above[13–15]. Anxiety symptoms alone increase the risk of cognitive impairment, future cognitive decline, and the development of clinical dementia[16,17]. Indeed, with each additional symptom of anxiety that an older adult develops, the risk of progressing from mild cognitive impairment to AD almost doubles (Relative 3-year risk=1.8/symptom)[18].

Therefore, any attempt to investigate the underlying neural basis of anxiety in aging can hold significant clinical and health benefits. Across the lifespan, at least two brain factors have been independently implicated in the development of anxiety. The first is related to structural changes in brain integrity, where both clinical and subclinical anxiety symptoms are linked

[1]Center for Human Sleep Science, Department of Psychology, University of California, Berkeley, California, USA. [2]Department of Psychology, University of California, Berkeley, California, USA. [3]Helen Wills Neuroscience Institute, University of California, Berkeley, California, USA. ✉e-mail: etibens@berkeley.edu; mpwalker@berkeley.edu

**Table 1 | Demographics, cognitive, and sleep summary information at visit 1(mean ± SD)**

| Variable | Mean ± SD |
|---|---|
| Age (years) | 74.57 ± 5.6 |
| Sex | 39 F/22 M |
| BMI | 25.25 ± 4.1 |
| AHI | 6.83 ± 8.8 |
| STAI Trait Anxiety | 28.61 ± 5.7 |
| Education (years) | 16.75 ± 1.87 |
| Total sleep time (min) | 337 ± 55.34 |
| Sleep Efficiency (%) | 70.58 ± 11.5 |
| NREM2 time (min) | 188.5 ± 52.52 |
| NREM3 time (min) | 64.15 ± 47.26 |
| REM time (min) | 49.56 ± 19.6 |
| WASO (min) | 111.14 ± 49.77 |
| MMSE | 29.11 ± 1.24 |

*BMI* Body Mass Index, *AHI* Apnea Hypopnea Index, *NREM* Non-rapid eye movement, *WASO* Wake after sleep onset, *SWA* slow wave activity, *MMSE* Mini-Mental State Examination.

with increased atrophy in regions that support emotion processing and regulation, including the Amygdala, Insula, Cingulate Cortex, and Para-hippocampal cortex[19–21]. Animal models further demonstrate significant morphological changes to limbic regions triggered by chronic stress[22], and similar indices of atrophy in humans are correlated with increased worry severity in late-life anxiety[20]. Moreover, patients with dementia demonstrate increased atrophy in limbic regions and lower glucose metabolism in the insula and parahippocampal regions if they suffer from anxiety, relative to dementia patients who do not[23–25].

The second factor consistently associated with the escalation and progression of anxiety states is poor sleep. This is true for clinical anxiety, wherein chronic sleep disruption more than doubles the risk of developing an anxiety disorder, as well as for subclinical anxiety states, which are elevated following nights of poor sleep[26–28]. Of the various sleep stages, robust literature has specifically linked NREM sleep disruption to the development and progression of anxiety. Reductions in NREM Slow Wave Sleep (SWS) and associated SWA have been reported in several anxiety disorders[29], including generalized anxiety disorder, PTSD[25,30], and subclinically in healthy individuals with high trait anxiety[31]. Demonstrating a beneficial link, greater NREM SWA (and associated SWS) in healthy individuals predict a greater anxiety reduction the next day, as well as increased activity in emotion regulation regions, such as the medial prefrontal cortex (mPFC), post-sleep[26]. Such findings posit a functional role for NREM SWA—an anxiolytic brain benefit preventing the overnight escalation of anxiety through prefrontal restoration of limbic regulation[32], thereby optimizing next-day affective regulation linked to lower anxiety states[33,34].

Such associations are relevant considering the independent and equally robust line of evidence linking impaired structural brain integrity with reductions in NREM SWA--both key hallmarks of the aging process[35–38]. Despite this evidence, whether age-related declines in SWA (i.e. SWA impairment) represent a significant factor accounting for anxiety changes across older adults remains uninvestigated and thus unknown. Moreover, if and how the underlying decline in SWA interacts with age-related structural brain deterioration and statistically accounts for escalating levels of next-day anxiety in older adults is similarly untested.

Motivated by recent evidence linking sleep, specifically NREM sleep, to overnight anxiety regulation in young adults, here we tested the hypothesis that anxiety symptoms in aging are in part related to co-occurring impairments in sleep compounded by the contribution of regional atrophy. Specifically, we tested the prediction that nighttime NREM SWA mediates the effect of atrophy on next-day anxiety. Such a finding would suggest that

increased atrophy hinders the creation of robust slow waves in older adults, thereby impairing overnight anxiety regulation.

In short (but see Methods), sixty-one cognitively normal older adults with a wide range of anxiety symptoms participated in the study (Table 1). Participants were given an 8-hour overnight EEG-recorded sleep opportunity, quantifying NREM oscillations across the scalp, together with validated measures of state anxiety before and after sleep. A structural MRI scan the following morning was further used to assess cortical and subcortical atrophy in anxiety-related regions.

## Methods
### Participants and experimental design

Seventy-four cognitively normal older adults were recruited from the Berkeley Aging Cohort Study. Exclusion criteria included history of neurologic, psychiatric, or sleep disorders, current use of antidepressant or hypnotic medications, and presence of contraindications for exposure to MRI imaging. The study was not preregistered and had been approved by the human studies committees of the University of California, Berkeley, and Lawrence Berkeley National Laboratory, with all participants providing written informed consent.

Participants spent two nights in the sleep lab, where they were given an 8-hour polysomnography (PSG) recorded sleep period, at a time that matched their habitual sleep-wake schedule, thus minimizing the influence of age-related circadian differences[39]. The first night served as an adaptation night to prevent first-night effects and allow for sleep disorders screening, whereas the second served as the experimental night. Following the experimental night, approximately 2 h post-awakening, structural MRI scans were obtained from all participants to quantify gray matter atrophy (more details below). Participants abstained from caffeine, alcohol, and daytime naps for 48 h before and during the experimental session. Moreover, participants kept a habitual sleep-wake profile for at least one week prior to the experimental session, validated using actigraphy and sleep logs.

Measures of anxiety were collected using the State-Trait Anxiety Inventory (STAI)[40], a 40-item Likert scale that assesses two dimensions of anxiety: state anxiety, reflecting transient feelings of anxiety (items 1–20), and trait anxiety, reflecting a more chronic profile of anxious behavior (items 21–40). All items are rated on a four-point Likert-like scale, resulting in a total score of 20 to 80 for either dimension, with higher scores indicating greater anxiety[41]. Trait measures were collected upon study enrollment, while state measures were assessed during the experimental session: twice in the evening before the sleep period (between 6 and 10 PM) and once in the morning following wake-up (between 7 and 9 AM). The measure of overnight anxiety change was calculated by subtracting the anxiety score post-sleep from the average of the two pre-sleep scores. Participants demonstrated a substantive variation in overnight anxiety change at baseline, with a similar number of participants reporting increases and decreases (including no change) in overnight anxiety (46% versus 49%, respectively).

Thirteen participants were excluded from the final analysis due to insufficient SWA detections ($N = 4$, see below), short sleep duration during the experimental night (<2 h, $N = 1$), or artifactual PSG recordings ($N = 3$). Participants were further excluded from the final analysis if they demonstrated extreme atrophy values in the chosen regions of interest (ROIs--see below, cutoff value of 2.5std below the sample mean, $N = 5$). The final sample, therefore, included 61 participants (Mean age=74.57 ± 5.6, 39 Females, 22 Males, 56 White, 4 Asian, sex and race based on self-report). Sample characteristics, sleep, and anxiety measures are summarized in Table 1. To quantify longitudinal changes in sleep and anxiety, a subset of participants ($N = 24$, age at follow-up=78.71 ± 5.9, 13 F/11 M) returned for a second visit to the sleep lab, with a mean follow-up duration of 4 ± 2.02 years. Measures of sleep and anxiety at the second visit were identical to the first.

### Sleep monitoring, SWA and spindle analysis

Whole-night PSG recording was collected using 19 electroencephalography (EEG) channels placed according to the 10–20 system, together with

**Table 2 | Brain volume and next-day anxiety (single ROI associations)**

| ROI | Laterality | Volume (Mean ± SD) | Overnight Anxiety Association (r) |
|---|---|---|---|
| Amygdala | Bilateral | 0.19 ± 0.026 | -0.22 (P = 0.1) |
| Insula | Left | 0.41 ± 0.034 | -0.11 (P = 0.42) |
| Putamen | Bilateral | 0.58 ± 0.083 | -0.27 (*P = 0.047) |
| Posterior Cingulate | Midline | 0.37 ± 0.045 | -0.23 (P = 0.095) |
| Parahippocampal Cortex | Left | 0.12 ± 0.02 | -0.03 (P = 0.83) |

*ROI volumes normalized for Total Intracranial Volume. * $P < 0.05$.

electrooculography (EOG) and electromyography (EMG). Reference electrodes were left and right mastoid (A1, A2), and data were digitized at 400 Hz using the Grass Technologies Comet XL system.

PSG data was scored using validated automatic sleep-scoring software[42] followed by visual inspection by two trained sleep-scoring professionals (Z.Z. and O.M.) based on standardized criteria[43]. In case of artifacts in one of the mastoid channels, EEG data were re-referenced from the contralateral mastoids to a unilateral mastoid (N = 5). During visual inspection, any EEG channel with marked artifact noise was flagged and omitted from subsequent analyses, including slow wave detections.

Following sleep scoring, slow waves (for both visits) were detected using a validated algorithm[42] with appropriate data-driven parameter adaptations for older adults[44]. Two key factors influenced the decision to focus on SW events rather than spectral power in the 1-4 Hz delta range: (1) Slow Wave events undergo several changes throughout healthy aging, demonstrating lower density and amplitude[45] as well as lower slopes and longer negative- and positive-phase durations[46]. Therefore, a focus on SW events provided a more nuanced examination of age-related changes in NREM sleep, rather than power alone. Moreover, (2) previous studies have shown an age-related decrease in spectral power in delta frequencies is more prominent in frontal derivations[47,48], whereas age-related reduction in SW density tends to be more constant over the scalp[45]. Such data further align with the experimental hypothesis targeting central derivations, which we have shown to be associated with anxiety regulation in young adults[26].

To detect Slow Wave events, data were first downsampled to 100 Hz and filtered between 0.5 and 2 Hz, considering the lower frequency of slow waves in older adults[45,46]. The following age-adapted parameters were then used to detect slow waves in each NREM2 and 3 epoch: peak-to-peak amplitude of 60 µV (negative amplitude 32 µV) and a wave duration of up to 2.5 seconds (see Figure. S1 for further parameters). Once all detections from a single participant were obtained, each wave was automatically examined for outlier values on frequency, amplitude, and duration parameters using anomaly-based isolation detection[49]. Outlier slow waves were subsequently removed from further analyses. To ensure an adequate sample of slow wave events, a minimum of 120 detections in Fz or Cz channels was required for each participant in the final analysis. Anxiety analyses focused, a-priori, on central channels (C3, C4, and Cz) based on previous work linking central SWA to clinical and non-clinical anxiety states[26,50]. The total sum of slow wave detections across central channels was calculated and log-transformed to allow for a more normal distribution before being implemented in the statistical analyses.

A similar pipeline was also used to detect spindle activity during baseline NREM sleep. Data were first downsampled to 100 Hz and filtered between 12 and 15 Hz. Since sleep spindles have been shown to have lower amplitude and duration in aging[51], the amplitude threshold for spindles was set to 2 standard deviations of the root mean square (RMS) of the filtered signal, and spindle duration was set to 0.4-2 s. Each candidate spindle needed to meet three additional criteria to be detected: (1) a correlation of 0.5 or above between the broadband filtered EEG signal (1–30 Hz) and the sigma filtered signal during spindle occurrence[52]; (2) the presence of at least one more spindle in a neighboring channel during the same time, and (3) a minimum of 5% relative power in the sigma band during a spindle. The goal of the latter is to make sure that the increase in sigma power is actually specific to the sigma frequency range and not due to a global increase in power (e.g., caused by artifacts). Spindles were detected when all 3 thresholds were met (see Figure. S2). Following detections, each spindle was automatically examined for outlier values on frequency, amplitude, and duration parameters, similar to the processing of slow waves described above[49]. Outlier spindles were subsequently removed from further analyses.

## Structural MRI analysis and atrophy assessment

Both clinical and subclinical anxiety symptoms are linked with increased atrophy in emotion-related brain regions such as the Amygdala, Insula, and Cingulate Cortex[19–21], a profile that is aggravated in patients with dementia[23–25]. Atrophy assessment, therefore, focused a priori on anxiety-sensitive ROIs, known to undergo significant changes in late-life anxiety[20,25], including the Amygdala, Posterior Cingulate, Putamen, left Parahippocampal cortex, and left Insula (see Table 2).

Structural MRI scans were obtained from all participants during their first experimental lab visit using high-resolution T1-weighted MPRAGE images (TR/TE = 1900/2.52 ms, FA = 9°, 1 × 1 × 1 mm resolution) collected via a Siemens Trio 3T scanner. The scans were processed using FreeSurfer version 5.3 (http://freesurfer.net/)[53–55]. After co-registering each image to MNI305[56], FreeSurfer reconstructs three-dimensional (3D) pial and white matter surfaces based on the relative intensity differences at white and gray matter tissue boundaries. Cortical parcellation is performed based on anatomical regions of interest defined by the Desikan-Killiany atlas[53,54]. The standardized FreeSurfer volumetric segmentation pipeline was then used to calculate volumes for each anatomical region as well as total intracranial volume[54]. To adjust for individual differences in head size, each anatomical region was normalized by total intracranial volume creating a normalized volume score across participants.

Volume assessment in anxiety-sensitive regions was then summed to create a single anxiety-related atrophy score per participant (i.e., single region ROIs were not used in the analyses. Non a-priori single ROI associations are presented in Table 2). Finally, a post hoc analysis explored the volume of seven standard brain networks using the same analysis steps described above to examine the specificity of anxiety-sensitive regions to overnight anxiety change. The networks were derived from a validated seven-network parcellation[57], including the limbic, frontoparietal, ventral/dorsal attention, somatomotor, default mode, and visual networks (see Table S1).

## Statistical analysis

Statistical analyses were conducted using the Pingouin library implemented in Python[58] and the JASP software (Version 0.18.3). Associations between the key variables of overnight anxiety, SWA, and atrophy were assessed using Pearson's correlation and multiple linear regression, including age, gender, and trait anxiety as covariates. Associations between anxiety and SWA at follow up were assessed using Spearman's correlation and paired t-tests to explore changes in SW detections across visits. All tests of statistical significance were two-sided, and p-values less than 0.05 were considered statistically significant. For each variable, data distribution plots were assessed for normality, and skewed variables were log-transformed to achieve normal distribution.

To test the hypothesis prediction that NREM SWA mediates the effect of regional atrophy on overnight anxiety, a formal mediation analysis was applied with regional atrophy as the independent variable, NREM SWA as the mediator variable, and overnight anxiety as the dependent variable. Specifically, the goal was to statistically determine whether NREM SWA could be deemed a mediator of the effect of regional atrophy on elevated anxiety states. The relevant outcome of formal mediation analysis is the indirect effect, which quantifies the difference between the effect of the independent variable on the dependent variable when the mediator is accounted for versus when it is not. Mediation analysis was performed using the mediation R package[59]. As

recommended for mediation analysis reporting[60], all effects were considered significant only if the 95% bias-corrected bootstrap confidence interval (of the indirect effect) was entirely above or below zero. Confidence intervals were derived from 1000 bootstrap samples, and the 95% confidence interval was computed by determining the indirect effects at the 2.5th and 97.5th percentiles.

## Results

### SWA in relation to overnight anxiety regulation

Testing the first hypothesis prediction, analyses sought to examine the role of SW events (0.5–2 Hz) in overnight anxiety regulation. Consistent with the experimental hypothesis, the greater the reduction in NREM slow waves, the greater the severity of anxiety the next day, an effect that was most significant in NREM2 sleep ($N = 58$, R = -0.38, 95% CI = [-0.58, -0.13], $P = 0.003$, Fig. 1). Importantly, these effects remained significant when controlling for age, gender, and trait anxiety levels ($\beta = -9.12$, 95% CI = [−15.58, −2.66], $R^2 = 0.16$, $P = 0.006$, see Table 3 for more details), as well as the addition of macro sleep features such as total sleep time, REM sleep amount, and sleep efficiency ($\beta = -9.17$, 95% CI = [−16.4345, −1.92], $R^2 = 0.18$, $P = 0.014$). Moreover, given that spindles dominate NREM2 sleep in addition to SWA, we further examined a potential involvement of spindle activity in overnight anxiety regulation. Analysis revealed no significant association between next-day anxiety and spindle counts, at either the predefined central derivation ($N = 58$, R = -0.23, 95% CI = [-0.46, 0.03], $P = 0.08$) or across the scalp (minimal $P = 0.5$, see Figure. S2 and Methods). Here too, the impact of SWA on overnight anxiety regulation remained significant when adding spindle count to the adjusted model together with age, gender, trait anxiety and the macro sleep features noted above ($\beta = -9.3$, 95% CI = [−16.54, −2.03], $R^2 = 0.198$, $P = 0.013$).

These findings suggest a selective benefit of SWA to overnight anxiety regulation beyond sleep duration, sleep quality, spindle activity, or REM sleep amounts, the latter a prominent feature in anxiety-related disorders such as PTSD[32,61]. Moreover, the prototypical beneficial impact of NREM SWA on anxiety appears to be preferentially linked to the regulation of daily fluctuations in anxiety states, separate from the stable metric of trait levels of innate anxiety. Further supporting this claim, exploratory analysis revealed no significant association between SWA at our predefined central derivative and trait anxiety across participants ($N = 58$, R = -0.16, 95% CI = [−0.4, 0.1], $P = 0.23$). That is, a state-dependent, day-to-day, sleep association with anxiety regulation, separate from trait. Last, despite the negative impact of aging on sleep, the role of sleep in anxiety regulation is not expressly age-dependent, fitting the anxiolytic benefit of SWA demonstrated in young adults[26].

These findings describe an association between sleep and next-day anxiety in healthy older adults, yet still leave open the question of the directional influence of this association. To estimate the unique contribution of sleep to anxiety regulation in older adults, we conducted a preliminary longitudinal analysis using a subset of participants who had a follow-up visit to the lab, which included both sleep and anxiety measures ($N = 24$, 13 F/ 11 M, mean follow-up 4 ± 2.02 years, see Methods for more details).

Findings from this analysis revealed the expected decline in NREM SWA with advanced age such that fewer SW events were detected during NREM sleep at follow-up relative to the first experimental visit (log SW amounts, visit 1 = 2.3 ± 0.24, visit 2 = 2.15 ± 0.36, t(23) = 3.43, 95% CI = [0.07, 0.29], $P = 0.002$; Figure. S3). Most importantly, this decrease in SW amounts was associated with increased anxiety levels at follow-up ($N = 23$, Rs= − 0.42, 95% CI = [−0.71, −0.01], $P = 0.04$; Figure. S3), affirming that the impact of deficient SWA on overnight anxiety regulation can causally and directionally instigate high levels of anxiety in healthy older adults, similar to findings in young adults[26,62]. Moreover, the longitudinal association between NREM SWA and next-day anxiety remained significant when controlling for baseline age, gender, and total sleep time ($\beta = -0.028$, 95% CI = [−0.05, 0], $R^2 = 0.29$, $P = 0.03$), suggesting that the causal impact of diminished SWA on anxiety is independent of age, gender and habitual sleep duration.

Such data highlight age-related SW disruption as a potential contributor to the development of late-life anxiety[63]. Notably, however, these findings do not invalidate the bidirectional interaction between sleep and anxiety, with studies demonstrating that daytime anxiety can trigger impairments in sleep[64,65], a phenomenon known as sleep reactivity[66]. Indeed, our findings are synergistic in this model. Specifically, sleep reactivity can create a negative feedback cycle wherein sleep disruption and escalating anxiety become self-reinforcing, contributing to the initial instigation of anxiety disorders, as well as their ongoing maintenance and/or worsening[28,67].

### Brain atrophy, SWA and overnight anxiety regulation

The next series of analyses tested the experimental prediction that increased atrophy in anxiety-sensitive regions (see Methods) results in greater next-day anxiety, rather than overnight anxiety decreases. Across participants, older age was significantly associated with increased atrophy in limbic and emotion-regulation regions, including the insula and amygdala (see Fig. 2 and Table S2). Moreover and fitting a neural signature of anxiety disorders[68], increased atrophy in anxiety-sensitive regions, was associated with greater levels of next-day anxiety ($N = 53$, R = -0.33, 95% CI= [−0.55, −0.06], $P = 0.016$, Fig. 2 and Table 2). This association also remained significant when controlling for age, gender, and trait anxiety levels ($\beta = -11.6$, 95% CI= [−22.89, −0.31], $R^2 = 0.16$, $P = 0.044$, see Table 3).

Thus, overnight changes in feelings of anxiety do not appear to be driven simply by age-related increases in atrophy or by long-lasting anxiety traits. Instead, increased atrophy in anxiety-sensitive regions appears to exert a dynamic impact on next-day anxiety, which is age and gender independent. Notably, these effects of atrophy on anxiety were specific to anxiety-sensitive regions. Exploring atrophy in other standard functional brain networks, including the default mode, somatomotor, and attentional networks, demonstrated no association with overnight anxiety change (minimal $P = 0.35$, FDR-corrected for multiple comparisons; see Methods and Table S1).

Consistent with the experimental hypothesis, increased atrophy in anxiety-sensitive regions was further associated with fewer NREM SW events across the night (N = 55, R = 0.46, 95% CI = [0.22, 0.65], $P < 0.001$, Fig. 2). Here, too, the association between SW and atrophy remained significant when controlling for age, gender, and trait anxiety levels ($\beta = 0.75$, 95% CI= [0.27, 1.21], $R^2 = 0.27$, $P = 0.002$, see Table 3), suggesting that the impact of atrophy on SW generation is not solely explained by age-related atrophy. Brain atrophy is age-related, but not age-dependent, and these findings build on that result, demonstrating that it is atrophy, more so than chronological age, that is most strongly associated with diminished SW events.

Since SWA impairment has previously been demonstrated to predict greater anxiety levels, the final analysis tested whether atrophy-associated reductions in SW amounts mediate higher levels of next-day anxiety in older adults, an outcome that was previously attributed to regional atrophy alone[14,20]. If correct, such a relationship would suggest that atrophy in anxiety-sensitive regions affects next-day anxiety by having an initial effect on nighttime SW generation that, in turn, impairs overnight anxiety regulation.

Supporting the prediction, a mediation model demonstrated that reduced SWA in older adults fully mediated the impact of atrophy on overnight anxiety regulation (bootstrapped unstandardized indirect effect $N = 53$, $\beta =-4.44$, 95% CI= [-9.36, -0.18], $P = 0.03$, Fig. 3). Therefore, rather than atrophy being directly associated with increased anxiety, instead, increased atrophy in anxiety-sensitive regions is related to impaired generation of SW events in older adults, and it is the severity of that SW impairment that, in turn, increases anxiety the next day (rather than the typical anxiolytic benefit of NREM SWA). Future examinations that target the experimental manipulation of SWA during sleep[69], in combination with anxiety measures, will aid in asserting the potential use of SWA enhancement as a factor that alters anxiety regulation in the face of age-related atrophy.

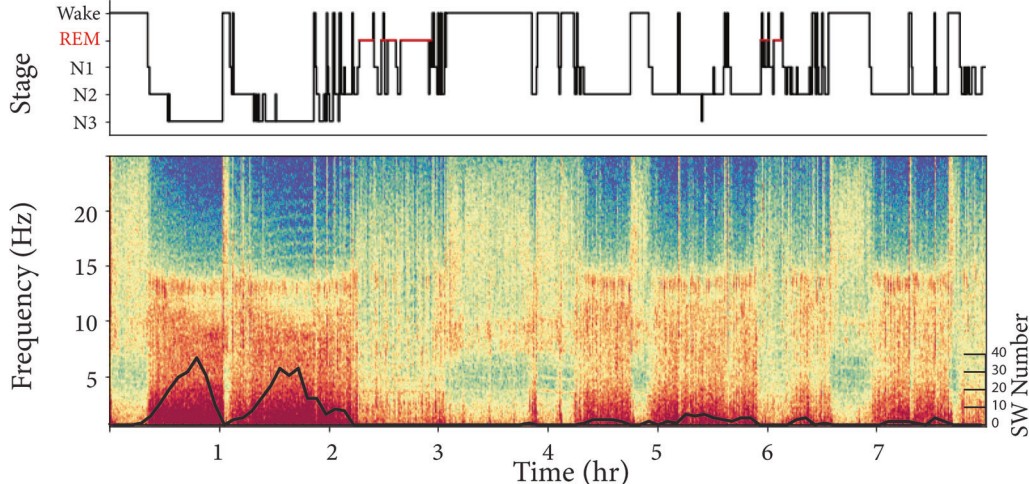

**A    Overnight Sleep (single subject)**

**B    NREM Slow Waves (0.5-2 Hz)**

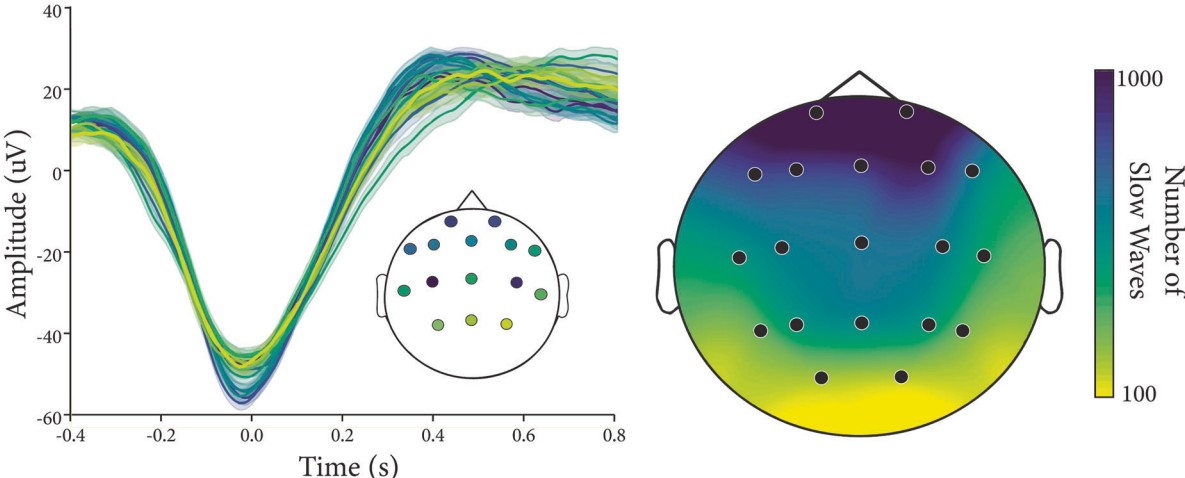

**C    Slow Waves and Overnight Anxiety Regulation**

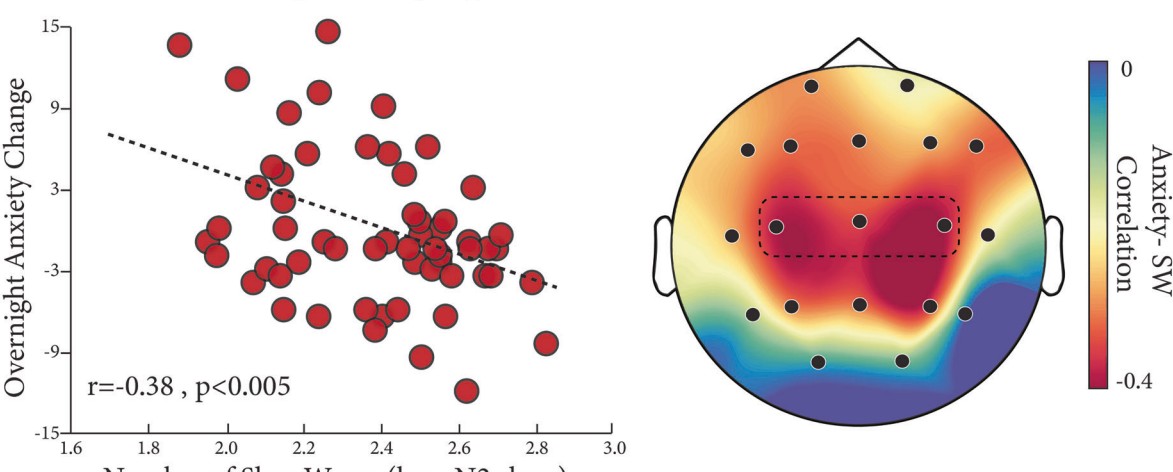

r=-0.38 , p<0.005

**Fig. 1 | NREM Slow Waves and Anxiety. A** Hypnogram of a single subject (top) with its corresponding multitaper spectrogram (Bottom, C3 channel). The super-imposed black line denotes the number of detected slow wave events across the night (5 min averages). **B** Detected slow waves from NREM sleep (left panel, single subject example, minimum 100 detections per plotted channel) and average slow wave counts across participants ($N = 61$, right panel). **C** Increased number of slow waves during NREM2 sleep is associated with lower next-day anxiety ($N = 58$, $R = -0.38$, $P < 0.005$, central derivative, left panel) also evident across the scalp (right panel, predefined central derivative marked in a dotted line, only channels with a minimum of 20 detections are plotted).

## Discussion

Collectively, these data support a framework in which impaired NREM SWA represents a previously under-recognized factor in regulating anxiety in older adults, potentially explaining the role of regional atrophy as an anxiety-promoting factor in late life[14,20,25]. At least two parsimonious, non-mutually exclusive pathways may underlie the consequences of aging-related SWA impairments on anxiety.

First is the otherwise beneficial shift in autonomic profile during NREM SWS from dominant sympathetic activity toward parasympathetic vagal tone. Such changes include SWA-dependent improvements in heart rate variability, suppression of hypothalamic–pituitary–adrenal axis, and a shift in neural network activity[70,71]. Indeed, SWA appears to significantly regulate affective brain networks during NREM, especially brain regions linked with autonomic control and emotion regulation[72], such as the cingulate cortex and the medial prefrontal cortex[73]. This association is also bidirectional. Across young and older adults, greater cortical thickness in the medial orbitofrontal cortex and anterior cingulate cortices is linked with higher NREM SWA, especially SWA in central derivations as reported here[37].

Through such central brain and peripheral sympathovagal regulation, when SWA is sufficient, it can aid in the overnight restoration of autonomic balance during NREM. This, in turn, leads to an overnight central and peripheral sympathetic regulation that promotes lower anxiety states[33,74,75]. In aging, however, when NREM SWA is disturbed, either by atrophy or pathological neurodegeneration, autonomic regulation is impaired. As a result of such sympathovagal imbalance, increased sympathetic dominance may lead to higher levels of next-day anxiety. This may therefore explain why, in part, sleep disturbances and dysfunctional arousal regulation are robustly linked to anxiety in community-dwelling older adults as well as in clinical cases of anxiety disorders, mild cognitive impairment, and AD[63,76,77].

A second pathway linking reduced SWA to anxiety regulation comes from recent evidence that NREM slow waves are associated with, and may provide an indirect biomarker of, declining central noradrenaline (NE) tone with advanced age. Studies in both rodents[78] and humans[79] have demonstrated that the activity of NE neurons in the Locus Coeruleus (LC) is temporally related to slow oscillations and spindle activity during NREM sleep[80-82], suggesting that LC and SWA integrity might fluctuate together in the context of aging. Notably, LC density, as assessed by high-field MRI, starts declining in cognitively unimpaired older individuals in

### Table 3 | Covariate results of the adjusted multiple linear regression models

| Covariate | SWA~ Overnight Anxiety | Overnight Anxiety~ Atrophy | SWA~Atrophy |
|---|---|---|---|
| Age | β = 0.028, P = 0.85 | β = 0.15, P = 0.34 | β = -0.006, P = 0.32 |
| Gender (F) | β = -0.04, P = 0.98 | β = 0.12, P = 0.93 | β = -0.009, P = 0.88 |
| Trait Anxiety | β = 0.02, P = 0.87 | β = 0.16, P = 0.2 | β = -0.007, P = 0.14 |

*SWA* Slow Wave Activity.

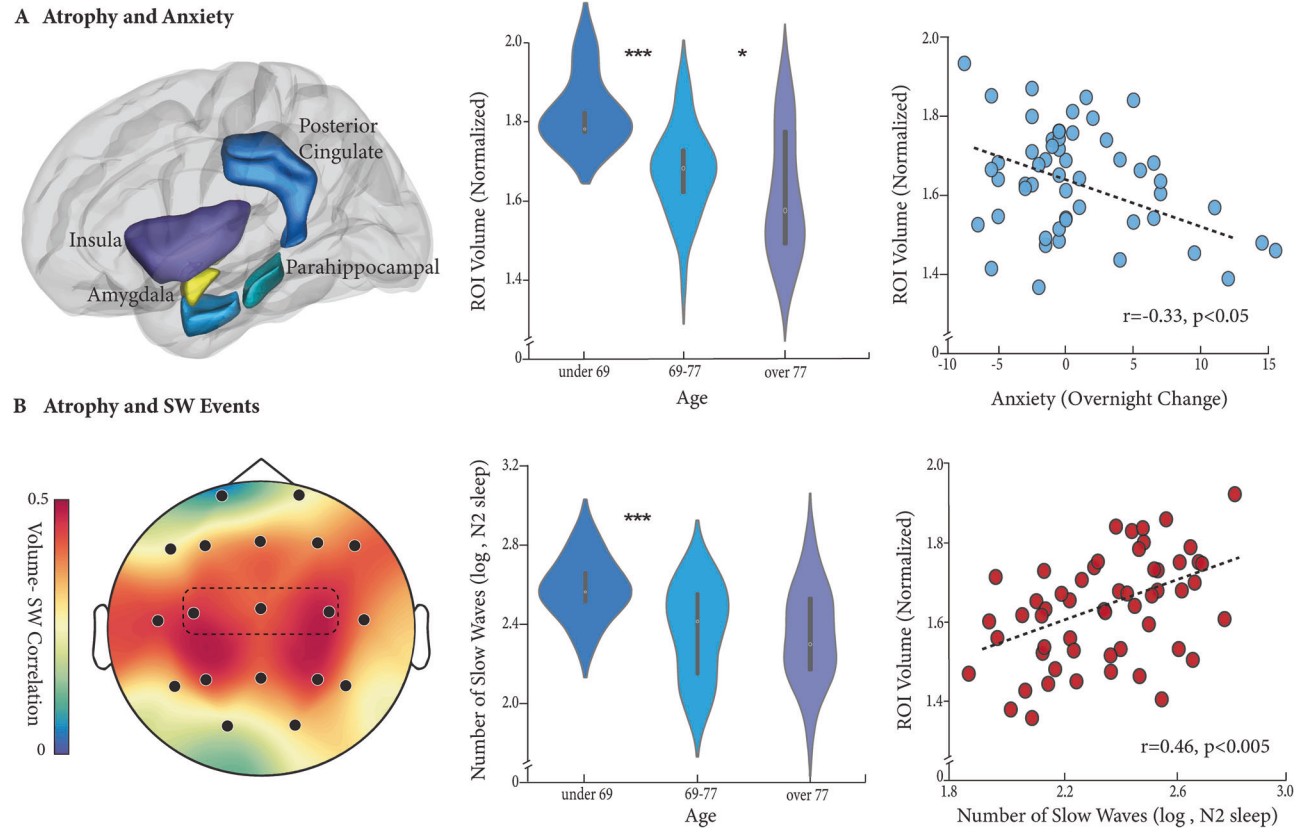

**Fig. 2 | Atrophy, NREM Slow Waves, and Anxiety. A** Anxiety-sensitive regions used to define regional atrophy across participants. These regions include the Amygdala (marked in yellow), Posterior Cingulate (blue), Putamen (anterior to the Insula), left Parahippocampal cortex (anterior-light blue, posterior-green), and left Insula (purple). Reduced volume in these regions was greater with increasing age (middle panel) and further associated with increased next-day anxiety (N = 53, R = -0.33, P < 0.05, right panel). *** P < 0.01, * P < 0.05. **B** Increased number of slow waves during NREM2 sleep was associated with higher volume in anxiety-sensitive regions (left panel, minimum 20 detections per channel, dotted line marks pre-defined central derivative) most evident in central scalp locations (right panel, N = 55, R = 0.46, P < 0.005, data from marked central derivative). In contrast, increased age is associated with fewer nighttime slow wave events (middle panel, NREM2 sleep, slow wave events measured from the same predefined central channel). *** P < 0.01.

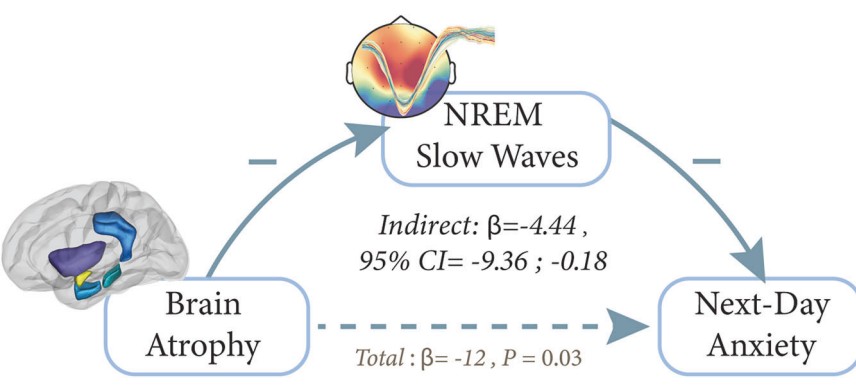

**Fig. 3 | The Mediating Role of Slow Waves on Overnight Anxiety Regulation.** Mediation analysis revealed a significant mediating effect of slow waves on anxiety regulation ($N = 53$, $P < 0.05$, bootstrapping simulations used to calculate 95% confidence intervals). Here, increased atrophy in anxiety-sensitive regions is linked to greater anxiety the next day through impaired slow wave generation/maintenance during NREM sleep.

the 5th decade of life[83] a time when SWA is significantly reduced[35,36]. Moreover, this LC deterioration is substantially greater in AD patients, proportionally linked with increasing amounts of Aβ and tau pathology, even in asymptomatic older individuals[84,85]. Therefore, impaired LC integrity may be one potent, though currently underappreciated, neurobiological pathway that explains the correspondence between impaired SWA and late-life anxiety in older adults. Parenthetically, abnormal LC function and increased physiological arousal are both key features in neurobiological explanations of late-life anxiety demonstrated in cases of PTSD, social anxiety disorder, and general anxiety disorder[14,86,87]. It may not, therefore, be coincidental that all these conditions are strongly comorbid with sleep disruption, including NREM sleep[26–28].

## Limitations

Our study had several limitations. First, anxiety was measured behaviorally using questionnaires rather than triggered by a controlled stress-inducing task, a choice made to avoid the known influence of evoked stress on subsequent sleep, allowing for a more ecological exploration of daily anxiety and SWA associations. Second, while the longitudinal subset provides preliminary directional evidence that deficient SWA can instigate elevated anxiety in healthy older adults, the cross-sectional nature of our baseline assessment limits strong causal inference; larger longitudinal cohorts will help clarify the temporal dynamics of the SWA–anxiety relationship. Third, we did not directly manipulate SWA through acoustic or transcranial stimulation, an intervention that would more definitively establish the clinical benefit of slow-wave enhancement for anxiety regulation. Fourth, our atrophy measures did not include biomarkers that are specific to Alzheimer's disease, such as amyloid-β or tau, leaving open the question of how preclinical pathology may interact with the SWA–anxiety pathway.

Despite these considerations, the convergence of cross-sectional, longitudinal, and mediation findings offers a coherent framework for understanding how impaired slow-wave sleep contributes to emotional dysregulation in aging, providing a foundation for next-step interventional assessments. Such assessments highlight a significant, but also actionable, benefit for emerging strategies that enhance NREM SWA, specifically in older adults[69,88,89]. Beyond the benefits to individuals and their caregivers of reducing the otherwise debilitating consequences of neuropsychiatric features, the potential to augment SWA in older adults may further help reduce the non-trivial healthcare costs associated with NPS features, helping to better manage mental health in aging populations[90].

## Data availability

Data used in this study is part of the Berkeley Aging Cohort and will be shared by application request from a qualified investigator at an academic institute, subject to the negotiation and decision of a university review and data use agreement process. Please contact EBS at etibens@berkeley.edu for further information. Data necessary to reproduce study figures is available at the OSF repository.

## Code availability

Codes and data necessary to reproduce study figures are available at the OSF repository.

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

## Acknowledgements

This work was supported by the National Institutes of Health (R01AG031164, RF1AG054019, and RF1AG054106 to M.P.W.), and funds from the University of California, Berkeley (M.P.W.). The funders had no role in study design, data collection and analysis, decision to publish or preparation of the manuscript.

## Author contributions

E.B.S. and M.P.W. conceptualized the study. V.D.S., O.M. and Z.Z. collected data. E.B.S., V.D.S., O.M. and Z.Z. analyzed the data. E.B.S. and M.P.W. wrote and edited the manuscript.

## Competing interests

The authors declare no competing interests.
