## [Transparent Peer Review file · Communications Psychology]

Impaired slow-wave sleep accounts for brain aging-related increases in anxiety

Corresponding Author: Dr Eti Ben Simon

Version 0:

Decision Letter:

Dear Dr Ben Simon,

Thank you for your patience during the peer-review process. Your manuscript titled "Slow to regulate: Impaired slow-wave sleep accounts for brain aging-related increases in anxiety" has now been seen by 2 reviewers, and I include their comments at the end of this message. They find your work of interest but raised some important points. We are interested in the possibility of publishing your study in Communications Psychology, but would like to consider your responses to these concerns and assess a revised manuscript before we make a final decision on publication.

We therefore invite you to revise and resubmit your manuscript, along with a point-by-point response to the reviewers. Please highlight all changes in the manuscript text file.

Editorially, we consider it crucial that the methodological concerns regarding the calculation of slow wave activity and the potential role of spindle activity are carefully addressed in the revised manuscript.

I am attaching an Editorial Requests Table that details critical reporting requirements for the revised manuscript. In particular, please make sure that the paper complies with the journal's formatting requirements as per the attached checklist. Please attend to each item and ensure your manuscript is fully compliant. If your revised manuscript is not aligned with these requests on major issues, such as those concerning statistics, it may be returned to you for further revisions without re-review.

Please submit the following items:

- Revised manuscript
- Point-by-point response to the referees' comments
- Cover letter (as a separate document)
- <https://www.nature.com/documents/nr-reporting-summary.pdf> Nature Research Reporting Summary
- Completed Editorial Request Table (attached).

via this link: Link Redacted .

Additional guidance is available in our style and formatting guide Communications Psychology formatting guide.

Best regards,
Trobey Lui, on behalf of

Xiaoqing Hu

Trobey Lui, PhD
Associate Editor
Communications Psychology

Xiaoqing Hu, PhD
Editorial Board Member
Communications Psychology
orcid.org/0000-0001-8112-9700

REVIEWER EXPERTISE:

Reviewer #1: sleep, aging

Reviewer #2: sleep, aging, mental health

REVIEWER REPORTS:

Reviewer #1 (Remarks to the Author):

This study examined the relationship between SWA and overnight anxiety in older adults. Results show that SWA was related to next-day anxiety, the age-related increase in anxiety was related to a decrease in SWA, and SWA mediated the relationship between brain atrophy and anxiety. The results definitely have high clinical importance. I only have a few questions.

1. Was SWA related to trait anxiety? Were changes in state anxiety overnight mostly observed in those with high trait anxiety?
2. Was there an age-related increase in trait anxiety at baseline, or at the longitudinal followup visit?
3. Why was the frequency range of 0.5-2Hz selected as SWA? Most studies have used variation between 0.5-1Hz, 0.8-4.5Hz etc. Can the authors provide a justification or reference for 0.5-2Hz?
4. The authors stated that "impairment in SWA" several times, but is there a clinical criterion for SWA impairment?

Reviewer #2 (Remarks to the Author):

Mental health and healthy cognitive aging are closely linked, highlighting the importance of investigating the neural basis of anxiety in older adults. Building on recent findings that associate slow wave activity (SWA) during NREM-sleep with overnight regulation of anxiety, Ben Simon and colleagues examined the link between SWA and anxiety in relation to atrophy in anxiety-specific regions. The main result showed that rather than atrophy being directly associated with increased post-sleep anxiety, greater atrophy in anxiety-sensitive regions was linked to decreased SW events which in turn linked to increased anxiety levels. While the results are compelling and the authors' interpretation is generally well-reasoned, I have some reservations regarding the specificity of the findings. These concerns could potentially be addressed through further clarification or additional analyses.

Major Comments:

- 1) Could you elaborate on the rationale for using slow-wave (SW) events rather than SW power? A higher number of SW events may reflect increased overall sleep, which you account for, but it could also correlate with higher spindle density or

even more REM sleep, which isn't controlled for.

1a) Given the well-established links between REM sleep and anxiety, I'm not fully convinced that SWA exerts specific effects independent of REM contributions.

1b) Additionally, since your first analysis showed the strongest effects in NREM2, it's worth considering whether these findings might be partially driven by spindle activity—particularly given your focus on central derivations, where spindles are prominently expressed.

2) In your longitudinal analysis, it's unclear whether you also controlled for trait anxiety and other relevant sleep variables such as total sleep time (TST). Clarifying this would strengthen the interpretation.

3) Regarding the mediation analysis, could you specify whether it was also conducted for other brain regions, as in the previous analysis? I'm particularly interested in frontal areas, given their prominent role in slow-wave generation.

Minor Comments:

1) In the abstract, '<65y' should likely be '>65y'

Version 1:

Decision Letter:

Dear Dr Ben Simon,

Your manuscript titled "Slow to regulate: Impaired slow-wave sleep accounts for brain aging-related increases in anxiety" has now been seen by our reviewers, whose comments appear below. In light of their advice I am delighted to say that we are happy, in principle, to publish a suitably revised version in Communications Psychology.

We therefore invite you to revise your paper one last time to address the remaining concerns of our reviewers and a list of editorial requests. At the same time we ask that you edit your manuscript to comply with our format requirements and to maximise the accessibility and therefore the impact of your work.

EDITORIAL REQUESTS:

SUBMISSION INFORMATION:

OPEN ACCESS:

Communications Psychology is a fully open access journal. Articles are made freely accessible on publication. For further information about article processing charges, open access funding, and advice and support from Nature Research, please visit <https://www.nature.com/commpsychol/open-access>

* DATA AVAILABILITY:

All Communications Psychology manuscripts must include a section titled "Data Availability" at the end of the Methods section. More information on this policy, is available in the Editorial Requests Table and at <http://www.nature.com/authors/policies/data/data-availability-statements-data-citations.pdf>

Link Redacted

Best regards,

Troy Lui, on behalf of

Xiaoqing Hu

Troy Lui, PhD
Associate Editor
Communications Psychology

Xiaoqing Hu, PhD
Editorial Board Member
Communications Psychology
orcid.org/0000-0001-8112-9700

REVIEWERS' COMMENTS:

Reviewer #1 (Remarks to the Author):

The authors have addresses my major concerns.

Reviewer #2 (Remarks to the Author):

The authors have addressed most of my concerns, and I am now more convinced that SWA appears to have a specific impact on next-day anxiety. I only have one final point regarding my initial comment 1:
Could the authors please indicate whether a model was conducted to test whether the association between NREM SWA and next-day anxiety remained significant when controlling for spindle count, similar to what was done with REM as a control variable?

AUTHOR'S RESPONSE IN BLUE TEXT

We are grateful for the reviewers' encouraging and thoughtful comments on our original submission, as well as those of the journal editors. Guided by these comments, we have now run an additional set of analyses, resulting in a substantively revised manuscript. Specifically, we were able to demonstrate the specificity and selectivity of the association between overnight anxiety and SWA, independent from their relationship to other sleep stages and oscillations, as suggested by the reviewers.

REVIEWER COMMENTS:

Reviewer #1 (Remarks to the Author):

This study examined the relationship between SWA and overnight anxiety in older adults. Results show that SWA was related to next-day anxiety, the age-related increase in anxiety was related to a decrease in SWA, and SWA mediated the relationship between brain atrophy and anxiety. The results definitely have high clinical importance. I only have a few questions.

Author's Response: Thanks so much for the encouraging comments. They really help.

1. Was SWA related to trait anxiety? Were changes in state anxiety overnight mostly observed in those with high trait anxiety?

Author's Response: While our findings demonstrate a significant association between SWA and overnight changes in anxiety, we found no such association with trait anxiety (N = 58, R = -0.16, P = 0.23). To further explore the impact of trait anxiety, we used a median split in the trait anxiety scores to create low (≤ 27 , N=31) and high (>27 , N=27) anxiety groups, as suggested by the reviewer. This split similarly revealed no significant difference between SWA ($t(59) = -0.1$, P = 0.92) or overnight anxiety change ($t(55) = -0.12$, P = 0.9) across participants with high and low trait anxiety. Such findings support the main study results and suggest that the beneficial impact of

NREM SWA on anxiety is preferentially linked to the regulation of daily fluctuations in anxiety states, separate from the stable metric of trait. These analyses were now added to the Results to highlight the state-dependent, day-to-day association between SWA and anxiety regulation, separate from trait. On pages 3-4, we now state:

“Moreover, the prototypical beneficial impact of NREM SWA on anxiety appears to be preferentially linked to the regulation of daily fluctuations in anxiety states, separate from the stable metric of trait levels of innate anxiety. Further supporting this claim, exploratory analysis revealed no significant association between SWA at our predefined central derivative and trait anxiety across participants (N = 58, R = -0.16, P = 0.23).”

2. Was there an age-related increase in trait anxiety at baseline, or at the longitudinal follow-up visit?

Author’s Response: This is an interesting suggestion. We examined whether age is associated with trait anxiety across both visits and found no significant association at either visit 1 (N = 58, R = 0.12, P = 0.36) or visit 2 (N = 24, R = 0.27, P = 0.19). We also examined whether trait anxiety differs across the same age groups that demonstrated an age-related change in atrophy and SWA in the main study (69 and younger, 69-77, and over 77, see Figure 2). An ANOVA analysis similarly revealed no significant association between these age groups and trait anxiety ($F(2, 55) = 0.167, P = 0.85$), suggesting that trait anxiety was not related to age in our study. We have now added these analyses to Supplementary Table 5 (see below), which also includes the association between age and atrophy/SWA as depicted in Figure 2. In the Methods (P. 12), we now state:

“Supporting prior research^{35,71}, older age was significantly associated with increased atrophy and fewer slow wave events during sleep at the baseline (see Fig. 2 and Table S5), with a continued reduction in NREM slow wave events at follow-up (see Fig. S2).”

Table S5. Age-based analysis of main study variables.

Covariate (Mean ± SD)	69 and younger	69-77	over 77	age-based ANOVA
----------------	-------	---------	--------------------

Trait Anxiety	28.17 ± 2.64	28.31 ± 5.58	29.19 ± 6.57	F(2, 55) = 0.167, P = 0.85
Volume in anxiety-sensitive regions	1.83 ± 0.08	1.69 ± 0.1	1.61 ± 0.15	F(2, 52) = 7.129, *P = 0.002
SWA (log count)	2.58 ± 0.15	2.37 ± 0.23	2.34 ± 0.22	F(2, 58) = 3.66, *P = 0.032

SWA = Slow Wave Activity, ROI volumes normalized for Total Intracranial Volume. * P < 0.05

3. Why was the frequency range of 0.5-2Hz selected as SWA? Most studies have used variation between 0.5-1Hz, 0.8-4.5Hz etc. Can the authors provide a justification or reference for 0.5-2Hz?

Author's Response: The reviewer is right to note the typical 1-4 Hz range used to calculate SWA, especially when exploring power in discrete frequency bands. Indeed, when we started our analyses, this was the range we used. However, there is a clear literature regarding age-related changes in sleep demonstrating a reduction in the typical frequency of individual slow waves (see, for instance, Carrier et al. 2011; Lafrenière et al 2023). Indeed, the majority of slow waves are centered on the 1Hz frequency range, leading us to adopt the age-adjusted SW criteria (Carrier et al. 2011; Lafrenière et al 2023) to ensure all SW events are correctly detected and accommodated for in older adults. We also note several studies demonstrating that SWA in the slower delta range (<2 Hz) is especially sensitive to the presence of Aβ and tau pathology in aging (e.g., Lucey et al., 2019; Mander et al., 2015), encouraging us to focus on a slower frequency range in our slow wave analysis. In addition to frequency, we also included amplitude adjustments (peak-to-peak amplitude of 60 μV, negative amplitude of 32 μV) and duration adjustments (up to 2.5 seconds) into the slow wave detection algorithm, guided by the findings of Rosinvil et al. (2021). To further clarify this point, we have now revised the Method section to state (P. 12):

“Following sleep scoring, slow waves (for both visits) were detected using a validated algorithm⁷² with appropriate data-driven parameter adaptations for older adults⁷⁴. Data were first downsampled to 100 Hz and filtered between 0.5 and 2 Hz, considering the lower frequency of slow waves in older adults (Carrier et al. 2011; Lafrenière et al 2023). The following age-adapted parameters were then used to detect slow waves in each

NREM2 and 3 epoch: peak-to-peak amplitude of 60 μ V (negative amplitude 32 μ V) and a wave duration of up to 2.5 seconds (see Fig. S2 for further parameters)."

4. The authors stated that "impairment in SWA" several times, but is there a clinical criterion for SWA impairment?

Author's Response: That's a good point. While macro-level sleep features, like sleep duration and efficiency, have recommended guidelines for older adults (e.g., Ohayon et al. 2017), micro-level oscillatory features such as SWA and/or spindles lack such guidelines and are typically assessed relative to younger age groups. That is, significant reductions in SWA are already observed in middle-aged adults relative to young adults, and this SWA impairment becomes especially prominent in older adults (e.g., Mander et al. 2017). To clarify this point, we have revised the manuscript to highlight that SWA impairment reflects an age-dependent decrease in SWA during aging. In the Introduction, we now state (P. 3):

"Despite this evidence, whether age-related declines in SWA (i.e., SWA impairments) represent a significant factor accounting for anxiety changes across older adults remains uninvestigated and thus unknown. Moreover, if and how the underlying decline in SWA interacts with age-related structural brain deterioration and statistically accounts for escalating levels of next-day anxiety in older adults is similarly untested."

In the Results, we state (P. 3):

"Testing the first hypothesis prediction, analyses sought to examine the role of SW events (0.5–2 Hz) in overnight anxiety regulation. Consistent with the experimental hypothesis, the greater the reduction in NREM slow waves, the greater the severity of anxiety the next day, an effect that was most significant in NREM2 sleep ($R = -0.38$, $P < 0.005$, Fig. 1)."

And on page 5:

"Since SWA impairment has previously been demonstrated to predict greater anxiety levels, the final analysis tested whether atrophy-associated reductions in SW amounts mediate higher levels of next-day anxiety in older adults, an outcome that was previously attributed to regional atrophy alone."

Reviewer #2 (Remarks to the Author):

Mental health and healthy cognitive aging are closely linked, highlighting the importance of investigating the neural basis of anxiety in older adults. Building on recent findings that associate slow wave activity (SWA) during NREM-sleep with overnight regulation of anxiety, Ben Simon and colleagues examined the link between SWA and anxiety in relation to atrophy in anxiety-specific regions. The main result showed that rather than atrophy being directly associated with increased post-sleep anxiety, greater atrophy in anxiety-sensitive regions was linked to decreased SW events which in turn linked to increased anxiety levels. While the results are compelling and the authors' interpretation is generally well-reasoned, I have some reservations regarding the specificity of the findings. These concerns could potentially be addressed through further clarification or additional analyses.

Major Comments:

1) Could you elaborate on the rationale for using slow-wave (SW) events rather than SW power? A higher number of SW events may reflect increased overall sleep, which you account for, but it could also correlate with higher spindle density or even more REM sleep, which isn't controlled for.

Author's Response: Thank you for raising this point. The choice to focus on SW events was motivated by a collection of findings establishing that: (1) Slow Wave events, more so than delta power, undergo marked changes throughout healthy aging, including lower slow-wave density and amplitude (Carrier et al. 2011), lower slopes and longer negative- and positive-phase durations (Lafrenière et al. 2023). Therefore, a focus on SW events provided a more nuanced examination of age-related changes in NREM sleep, rather than power alone, and (2) Previous studies have shown that an age-related decrease in spectral power in delta frequencies is more prominent in frontal derivations (e.g., Munch et al., 2004; Robillard et al., 2010), whereas age-related reduction in SW density tends to be more constant over the scalp (Carrier et al. 2011). Such data further align with the experimental hypothesis targeting central derivations, which we have shown to be associated with anxiety regulation in young adults (Ben Simon et al. 2020). To further emphasize this point, we revised the Methods (P. 12) to state:

“Following sleep scoring, slow waves (for both visits) were detected using a validated algorithm⁷² with appropriate data-driven parameter adaptations for older adults⁷⁴. Two key factors influenced the decision to focus on SW events rather than spectral power in the delta (1-4Hz) range: (1) Slow Wave events undergo several changes throughout healthy aging, demonstrating lower density and amplitude (Carrier et al. 2011) as well as lower

slopes and longer negative- and positive-phase durations (Lafrenière et al. 2023). Therefore, a focus on SW events provided a more nuanced examination of age-related changes in NREM sleep, rather than power alone, and (2) Previous studies have shown that an age-related decrease in spectral power in delta frequencies is more prominent in frontal derivations (Munch et al., 2004; Robillard et al., 2010), whereas age-related reduction in SW density tends to be more constant over the scalp (Carrier et al. 2011). Such data further align with the experimental hypothesis targeting central derivations, which we have shown to be associated with anxiety regulation in young adults (Ben Simon et al. 2020)."

1a) Given the well-established links between REM sleep and anxiety, I'm not fully convinced that SWA exerts specific effects independent of REM contributions.

Author's Response: The reviewer importantly points out that REM sleep can contribute to overnight anxiety regulation in addition to SWA. Disturbed REM sleep is strongly linked with anxiety disorders such as PTSD, and we agree that its contribution should be accounted for. It is relevant to note, however, that NREM disruptions are just as common in anxiety disorders as REM disruptions (e.g., Richrads et al. 2020), the former reported in generalized anxiety disorder, panic disorders (especially as triggers to nocturnal panic attacks) and in PTSD, suggesting that the signature impairment in NREM is trans-diagnostic and common across several clinical and subclinical anxiety states. Still, to further corroborate the specific contribution of SWA to overnight anxiety regulation, we conducted a series of analyses that focused on REM sleep. First, we examined whether REM sleep is linked with increased SW events as suggested by the reviewer. Analysis revealed that indeed spending more time in REM sleep is linked with greater SW events in NREM 2 (N = 61, R = 0.29, P = 0.02), likely reflecting overall increased sleep duration as noted by the reviewer. We therefore added REM sleep amounts to the main model that examines the link between SWA and overnight anxiety regulation. This analysis revealed that the association between NREM SWA and next-day anxiety remains significant when controlling for REM sleep amount ($\beta = -9.19$, 95% CI = [-16.45, -1.92], $R^2 = 0.18$, P = 0.01), suggesting a unique contribution of SW events to overnight anxiety regulation, independent of REM sleep. This analysis was now added to the manuscript. On page 3, we write:

"Moreover, the association of NREM SWA with next-day anxiety remained significant when controlling for macro sleep features such as total sleep time, REM sleep amount,

and sleep efficiency ($\beta = -9.19$, 95% CI = $[-16.45, -1.92]$, $R^2 = 0.18$, $P = 0.01$)... These findings suggest a selective benefit of SWA to overnight anxiety regulation beyond sleep duration or quality, spindle activity, or REM sleep amounts, the latter a prominent feature in anxiety-related disorders such as PTSD (Richrads et al. 2020)”.

1b) Additionally, since your first analysis showed the strongest effects in NREM2, it's worth considering whether these findings might be partially driven by spindle activity—particularly given your focus on central derivations, where spindles are prominently expressed.

Author's Response: This is an important point. Spindles are dominant in NREM2 sleep, and the central location of our SWA findings also suggests a potential involvement of spindle activity in overnight anxiety regulation. It may therefore be that sleep-associated changes in spindle activity explain the reported changes in overnight anxiety, rather than SWA alone. We have now conducted a series of new analyses to fully examine the specificity of our anxiety associations with SWA, relative to changes in spindle activity. First, we detected spindles using a validated algorithm applied to the same N2 and N3 epochs used to detect SW across all participants and scalp derivations (see more details below). Second, spindle counts were log-transformed and averaged across the same central locations used to explore SW events and overnight anxiety associations. We then examined the association between spindle amounts and overnight anxiety regulation during NREM2. Findings revealed that despite an overall trend towards reduced next-day anxiety with increased spindle counts, the association between spindle activity and overnight anxiety regulation was not significant at either our predefined central derivation (N=58, $R = -0.23$, $P = 0.08$) or across the scalp (all channels $P > 0.5$, see Figure S1 below). This data indicates that despite the co-occurrence of spindles and SW events in NREM2, it is SWA that contributes to nightly anxiety regulation. Further supporting this claim, atrophy in anxiety-sensitive regions showed no association with spindle activity (N=55, $R = 0.09$, $P = 0.53$), suggesting that atrophy in these regions does not contribute to the generation or propagation of spindle activity, as opposed to its impact on SWA. To address this topic adequately in the manuscript, we have revised the Introduction, Methods, and Supplementary Information.

In the Results, we now state (P. 3):

“Consistent with the experimental hypothesis, the greater the reduction in NREM slow waves, the greater the severity of anxiety the next day, an effect that was most significant in NREM2 sleep ($R = -0.38$, $P < 0.005$, Fig. 1). Importantly, these effects remained significant when controlling for age, gender, and trait anxiety levels ($\beta = -9.13$, 95% CI =

[-15.61, -2.66], $R^2= 0.16$, $P < 0.01$, see Table S2 for more details). Moreover, the association of NREM SWA with next-day anxiety remained significant when controlling for macro sleep features such as total sleep time, REM sleep amount, and sleep efficiency (-8.31, 95% CI = [-15.44, -1.2], $R^2= 0.15$, $P < 0.05$). Given that spindles dominate NREM2 sleep in addition to SWA, we further examined a potential involvement of spindle activity in overnight anxiety regulation. Analysis revealed no significant association between next-day anxiety and spindle counts, at either the predefined central derivation (N=58, $R = -0.23$, $P = 0.08$) or across the scalp (all channels $P > 0.5$, see Fig. S1 and Methods for more details). These findings suggest a selective benefit of SWA to overnight anxiety regulation beyond sleep duration or quality, spindle activity, or REM sleep amounts, the latter a prominent feature in anxiety-related disorders such as PTSD (Richrads et al. 2020)”.

In the Methods, we now describe the spindle detection process (see **Sleep monitoring, spindles, and SWA analysis** on P. 13):

“A similar pipeline was later used to detect spindle activity during NREM sleep. Data were first downsampled to 100 Hz and filtered between 12 and 15 Hz. Since sleep spindles have been shown to have lower amplitude and duration in aging (e.g., Martin et al. 2013), the amplitude threshold for spindles was set to 2 standard deviations of the root mean square (RMS) of the filtered signal, and spindle duration was set to 0.4-2 seconds. Each candidate spindle needed to meet three additional criteria to be detected: 1) a correlation of 0.5 or above between the broadband filtered EEG signal (1-30 Hz) and the sigma filtered signal during spindle occurrence (Lacourse et al. 2018); 2) the presence of at least one more spindle in a neighboring channel during the same time and 3) at least 5% relative power in the sigma band during a spindle. The goal of the latter is to make sure that the increase in sigma power is actually specific to the sigma frequency range and not just due to a global increase in power (e.g., caused by artefacts). Spindles were detected when all 3 thresholds were met. Following detections, each spindle was automatically examined for outlier values on frequency, amplitude, and duration parameters, similar to the processing of slow waves described above⁸⁰. Outlier spindles were subsequently removed from further analyses.”

And in the Supplementary Information, we added Figure **S1** to describe the results (P. 20):

A NREM Spindles (12-15 Hz)

B Spindle Activity and Overnight Anxiety Regulation

Fig. S1. NREM Spindles and Anxiety.

(A) Detected spindles from NREM sleep (left panel, single subject example) and average spindle counts across participants (right panel).

(B) Increased number of spindles during NREM2 sleep was not significantly associated with lower next-day anxiety ($R=-0.23$, $P = 0.08$, left panel) in either a predefined central derivative (right panel, marked in a dotted line) or across the scalp (right panel).

2) In your longitudinal analysis, it's unclear whether you also controlled for trait anxiety and other relevant sleep variables such as total sleep time (TST). Clarifying this would strengthen the interpretation.

Author's Response: Thank you for pointing out this topic. The original analysis did not include covariates and focused on the longitudinal relationship between SW events and anxiety across study visits. We have now reanalyzed the longitudinal data using a model that controls for key

metrics, including age and gender (at baseline), as well as baseline total sleep time, as suggested by the reviewer. Findings reveal that the longitudinal association between NREM SWA and next-day anxiety remains significant when controlling for these covariates ($\beta = -0.02$, 95% CI = [-0.04, 0], $R^2 = 0.15$, $P < 0.05$), affirming that the impact of reduced SWA on anxiety regulation can causally and directionally instigate high levels of anxiety in healthy older adults, independent of age, gender and habitual sleep duration. This analysis was now added to the manuscript. On page 4, we write:

“Most importantly, this decrease in SW amounts was associated with increased anxiety levels at follow-up ($R_s = -0.42$, $P = 0.04$; Fig. S1), affirming that the impact of deficient SWA on overnight anxiety regulation can causally and directionally instigate high levels of anxiety in healthy older adults, similar to findings in young adults^{26,39}. Moreover, the longitudinal association between NREM SWA and next-day anxiety remained significant when controlling for baseline age, gender, and total sleep time ($\beta = -0.02$, 95% CI = [-0.04, 0], $R^2 = 0.15$, $P < 0.05$), suggesting that the causal impact of diminished SWA on anxiety is independent of age, gender and habitual sleep duration.”

3) Regarding the mediation analysis, could you specify whether it was also conducted for other brain regions, as in the previous analysis? I’m particularly interested in frontal areas, given their prominent role in slow-wave generation.

Author’s Response: The mediation analysis focused on anxiety-sensitive regions, given our main hypothesis linking SWA to overnight anxiety regulation. As mentioned in the results, atrophy in other functional networks, including the frontoparietal network, was not predictive of overnight anxiety changes and was therefore not included in the mediation analysis. For descriptive purposes, we have now added Supplementary Table 4 (see below) to describe the specific association of each functional network with both SWA and overnight anxiety regulation at visit 1.

Table S4. Brain volume of key functional networks and next-day anxiety associations

Functional Network	Volume (mean \pm SD)	Overnight Anxiety Association (r)	SWA Association (r)
------------------------	-----------------------------------	---------------------

Visual	4.17 ± 0.41	-0.18 (P = 0.2)	0.19 (P = 0.16)
Somatomotor	3.99 ± 0.34	-0.07 (P = 0.6)	0.18 (P = 0.17)
Attention (dorsal)	2.72 ± 0.22	-0.13 (P = 0.36)	0.28 (*P = 0.04)
Attention (ventral)	2.87 ± 0.23	-0.22 (P = 0.11)	0.25 (P = 0.06)
Limbic	2.95 ± 0.27	-0.09 (P = 0.5)	0.17 (P = 0.21)
Frontoparietal	3.75 ± 0.3	-0.17 (P = 0.2)	0.26 (P = 0.056)
Default Mode	6.42 ± 0.5	-0.2 (P = 0.15)	0.33 (*P = 0.01)

Minor Comments:

1) In the abstract, '<65y' should likely be '>65y'

Author's Response: Revised. Thanks for catching that!